# Dual Closed-Loops Capacity Evolution Prediction for Energy Storage Batteries Integrated with Coupled Electrochemical Model

**Bowen Xu** [1,2]**, Tao Sun** [1]**, Shuoqi Wang** [2]**, Yifan Wei** [2]**, Xuebing Han** [2] **and Yuejiu Zheng** [1,2,*]

1   College of Mechanical Engineering, University of Shanghai for Science and Technology, Shanghai 200093, China; bwx97auto@163.com (B.X.); jd_rs7@163.com (T.S.)
2   State Key Laboratory of Automotive Safety and Energy, Tsinghua University, Beijing 100084, China; 15295517398@163.com (S.W.); axioncaoshen@163.com (Y.W.); jiaxp1997@163.com (X.H.)
*   Correspondence: yuejiu_zheng@163.com

**Abstract:** The health assessment for energy storage batteries matters in the context of carbon neutrality. Dual closed-loops capacity framework integrated with a reduced-order electrochemical model including triple side reactions is put forward, realizing parameter correction for health evaluation. Simplified microgrid aging experiment is formulated to test the closed-loop matching between the aging mechanism and electrochemical model relying on incremental capacity analysis. In addition, taking into account the future degradation prediction for energy storage system, the reliable capacity output afterwards acts as references for closed-loop parameter updating in empirical model to predict degradation evolution. The framework proposed implements the closed-loop dynamic updating for aging parameters with ideal error within 2%, making up for the lack of aging mechanism interpretation of accustomed empirical or data-driven black box model in the field of energy storage batteries.

**Keywords:** energy storage; state of health; health diagnosis; life prediction; electrochemical model

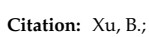



## 1. Introduction

The microgrid dominated by renewable energy for electricity generation is believed to be a potential solution to alleviate the increasing regional charging burden of electric vehicles (EVs). More service providers are bundling EVs charging stations with microgrids to gain power from nature. Considering that both the photovoltaics (PV) and wind turbines suffer from randomness and volatility, the battery energy storage system (ESS) thence plays a vital role in the microgrid to smooth the fluctuation of power generation, supplying a stable energy supply [1]. Graphite-LiFePO4 battery is deemed to be the most promising candidate material for large-scale energy storage thanks to its advantages of ideal chemical and thermal stability [2]. During the configuration of graphite-LiFePO4 battery ESS, taking its relatively large economic cost into account, one of the most important issues is to determine its optimal capacity. The occurrence of battery degradation causes the aging model to be considered in the capacity optimization, so that the cycling performance of ESS can be modeled and accurately evaluated during the full life [3]. Therefore, the establishment of a battery aging model under microgrid operations really matters. In fact, due to the recent popularity of EVs, the degradation of lithium-ion batteries used as vehicle power has been extensively and in-depth studied compared to renewable energy storage [4,5]. Distinct from the working conditions, there is a certain connection between the above two aspects due to similar objects.

Various technical routes have been developed to evaluate the state of health (SOH) of graphite-LiFePO4 batteries. Battery aging models for energy storage conditions can generally be divided into empirical-based and electrochemical-based model. The empirical degradation model is established with dynamic conditions [6], which takes the influence of

state of charge (SOC), C rate and Ah throughput into account. Although the prediction error is controlled ideally, the actual application accuracy fails to be guaranteed as the operating conditions change since the aging mechanism is not involved in the empirical model, bringing about the parameter mismatch of the empirical model [7].

The electrochemical model has recently been advanced with the ability to describe the internal side reaction mechanism inside, which is able to more accurately calculate the capacity loss. Reference [8] derives the coupling electrochemical-thermomechanical aging model relies on the pseudo two-dimension (P2D) model, and gives the analysis of various states within the battery. However, its calculation cost is too expensive for energy storage conditions with long operating cycles. In addition, the difficulty in accurately obtaining various parameters in P2D models also hinders the application [9]. To balance the trade-off between authenticity and computational cost, the reduced-order physical aging model represents a feasible solution for health assessment under microgrid energy storage conditions. Based on the physical degradation mechanism, the reduced-order capacity loss model of the graphite-LiFePO4 battery is proposed in [10], yet has the effectiveness with actual microgrid operating conditions not been verified. As far as we know, the physically-based reduced-order battery aging model with energy storage conditions has not been deduced or verified by experiments.

To make up for this research gap, this paper brings forward a coupled reduced-order degradation model for energy storage batteries, in which the side reactions include solid electrolyte interface (SEI) layer growth, the loss of active materials (LAM) and lithium plating. To verify the practical applicability, the cycling aging experiment is devised from the actual operating conditions of the microgrid, so the application of the degradation model can be well coordinated with the actual application in practice. During the verification, the qualitative incremental capacity analysis is employed to verify whether the aging mechanism matches the model calculation, forming the first closed loop: mechanism interpretation. On the other hand, the dominant aging models applied are open-loop, which means that the parameters fail to be updated during the lifespan [7]. In fact, parameter updating indeed matters to avoid mismatches. To overcome the aforementioned issues, another closed loop is proposed: the capacity output of the degradation model above acts as a reference to update the aging parameters in a closed loop, so as to realize the parameter correction during the whole life cycle. So far, the core dual closed-loops capacity prediction framework of this paper has been wholly illustrated.

The paper chiefly sets up a coupled reduced-order degradation model for actual energy storage. Subsequently, the reliable model output supplies observations for the parameter update of the empirical model for degradation evolution prediction. The remainder of the paper is organized as follows: Section 2 introduces the triple sub-models of the reduced-order physical degradation model. The model calibration and experimental verification are given in Section 3, while the dual closed-loops capacity prediction framework is expounded in Section 4. Finally, the conclusion is given in Section 5.

## 2. Reduced-Order Physical Degradation Model

To assess battery health under energy storage conditions, the battery degradation model ought to compromise between authenticity and computational efficiency. As a common type of reduced-order model, the single particle (SP) model is adopted. The anode and cathode are assumed to be two single particles as shown in Figure 1, with its radius $R_n$ and $R_p$ respectively. The reduced-order degradation model for energy storage proposed in this paper is derived from the SP model with three specific physical degradation side reactions included. Fundamentally speaking, the reduced-order degradation model is a semi-empirical model considering the aging mechanism, which is calibrated relying on cycling life tests.

**Figure 1.** Schematic diagram of reduced-order sp degradation model.

*2.1. SEI Growth Sub-Model*

Assuming that the growth of SEI is irreversible, numerous studies based on the Tafel equation have been carried out to model the growth dynamics of SEI [11]. Taking the influence of solvent diffusion and reduction kinetics on the growth of SEI into account, the SEI formation rate $i_{SEI}$ is solved by the solvent concentration on the anode surface $C_s(R_n, t)$ and the overpotential of the side reaction $\eta_{SEI}$ [12]

$$i_{SEI}(t) = nFk_{SEI}C_s(R_n, t)\exp\left(-\frac{\alpha F}{RT_{bat}}\eta_{SEI}\right) \tag{1}$$

where $n$ denotes the number of electrons reduced in the reaction, $F$ is the Faraday constant and $k_{SEI}$ is the rate constant of the SEI reaction. $\alpha$ and $R$ represent the transfer coefficient and ideal gas constant, respectively. In addition, $C_s(R_n, t)$ is determined by Fick's second diffusion law and subjected to the initial and boundary conditions [13]. The capacity loss caused by the SEI formula can be quantified as

$$Q_{loss} = \int_0^T i_{SEI}(t)dt \tag{2}$$

The SEI formation rate $i_{SEI}$ can be obtained relying on the Laplace transform and the Nernst equation detailed in [10] in the equilibrium state, as shown in Equation (3)

$$i_{SEI}(t) = \frac{k_{SEI}}{2(1+\lambda\theta)\sqrt{t}}\exp\left(-\frac{E_{SEI}}{RT}\right) \tag{3}$$

where $\lambda$, $k_{SEI}$ and $E_{SEI}$ are parameters calibrated according to cycling life tests, $T$ is the battery temperature, and $\theta$ is solved as

$$\theta = \exp\left(\frac{nF}{RT}\eta_{SEI}\right) = \exp\left[\frac{nF}{RT}\left(\eta_m + U_n^{ref} - U_{SEI}^{ref}\right)\right] \tag{4}$$

The formula is further explained, the equilibrium potential of the anode $U_n^{ref}$ is a function of SOC [13]. Simultaneously $U_{SEI}^{ref}$ denotes the open circuit potential of SEI growth. Besides, the overpotential of the main reaction $\eta_m$ is calculated according to Equations (5) and (6).

$$\eta_m = \frac{RT}{\alpha F}ln\left(\xi + \sqrt{1+\xi^2}\right) \tag{5}$$

$$\xi = \frac{R_n I}{6i_0 \varepsilon_{s,n0} A \delta_n} \tag{6}$$

where $I$ is the charge/discharge current, $i_0$ is the exchange current density, and $\varepsilon_{s,n0}$ is the initial volume volume fraction of the solid particle of the anode. $A$ and $\delta_n$ are the active specific surface area and anode thickness, respectively. It is worth noting that the SEI film thickness $\delta_{SEI}$ and resistance $R_{SEI}$ both increase with the side reaction of SEI film growth and the growth rate could be obtained according to $i_{SEI}$ in Equations (7) and (8). Where $M$, $\rho$, and $\sigma_{SEI}$ are the molecular weight, density and conductivity of the SEI layer, respectively. The parameters above $\delta_{SEI}$ and $R_{SEI}$ are significant in the coupled and physical degradation model, influencing the operation of other sub-models.

$$\frac{\partial \delta_{SEI}}{\partial t} = \frac{i_{SEI} M}{\rho n F} \tag{7}$$

$$\frac{\partial R_{SEI}}{\partial t} = \frac{1}{\sigma_{SEI}} \frac{\partial \delta_{SEI}}{\partial t} \tag{8}$$

*2.2. LAM Sub-Model*

Based on Miner's rule, the LAM model caused by mechanical stress has been researched and developed. Supposing that each cycle contributes to the cumulative fracture of the active material, the LAM process can be simulated relying on Equation (9).

$$i_{LAM}(t) = k_{LAM} \left( \frac{\sigma_{h,max}(R_n) - \sigma_{h,min}(R_n)}{\sigma_{yield}} \right)^{\frac{1}{m}} \tag{9}$$

where $k_{LAM}$, $\sigma_{yield}$, and $m$ require to be calibrated according to the cycling life tests. $\sigma_{h,max}$ and $\sigma_{h,min}$ are respectively the maximum and minimum hydrostatic stress in the measurement cycle, and the hydrostatic stress $\sigma_h$ is given as

$$\sigma_h(R_n) = \frac{\sigma_r(R_n) + 2\sigma_t(R_n)}{3} \tag{10}$$

$\sigma_h(R_n)$ and $\sigma_t(R_n)$ respectively represent the radial and tangential stresses of the spherical particles at the radius $R_n$. The two variables can be calculated through the core–shell stress–strain model as shown in Equation (11).

$$\begin{cases} \sigma_r(R_n) = -\frac{2E_p}{9(1-v_p)} \int_0^{c_p} \Omega_p(c_p) dc_p + \frac{a_p E_p}{1-2v_p} - \frac{2b_p E_p}{R_n^3(1+v_p)} \\ \sigma_t(R_n) = \frac{E_p}{9(1-v_p)} \int_0^{c_p} \Omega_p(c_p) dc_p + \frac{a_p E_p}{1-2v_p} + \frac{b_p E_p}{R_n^3(1+v_p)} - \frac{\Omega_p(c_{p,max}) E_p c_{p,max}}{3(1-v_p)} \end{cases} \tag{11}$$

where $\Omega_p$ is the partial molar volume related to the anode lithium concentration $c_p$. $E_p$ and $v_p$ denote respectively the Young's modulus and the Poisson's ratio of the graphite particle. $a_p$ and $b_p$ are decoupled through solving the linear equation below.

$$\begin{cases} a_p - a_s - \frac{b_s}{R_n^3} = -\frac{1+v_p}{9(1-v_p)} \int_0^{c_p} \Omega_p(c_p) dc_p \\ \frac{a_s}{1-2v_s} = \frac{2b_s}{R_s^3(1+v_s)} \\ -\frac{E_p a_p}{1-2v_p} + \frac{E_s a_s}{1-2v_s} - \frac{2E_s b_s}{R_p^3(1+v_s)} = -\frac{2E_p}{9(1-v_p)} \int_0^{c_p} \Omega_p(c_p) dc_p \end{cases} \tag{12}$$

The radius, Young's modulus, and Poisson's ratio of SEI film layer are represented as $R_s$, $E_s$, and $v_s$, respectively. It is obvious in Figure 1 that the radius of the SEI film layer $R_s$ is the sum of the radius of the spherical particles $R_n$ and the thickness of the SEI film

$\delta_{SEI}$. $\int_0^{c_p} \Omega_p(c_p)dc_p$ is the graphite volume change for lithiation and delithiation, which could be approximated as a function of lithiation degree or SOC. Complicated integral calculation can be omitted in this way as to realize the acceleration for calculation, which enables the evaluation of health status in a long period to be carried out. The relationship between the volume change of graphite and SOC is obtained in [14]. So far, the proposed LAM sub-model has been interpreted allowedly.

### 2.3. Lithium Plating Sub-Model

Lithium plating is one of the most serious side reactions for lithium-ion batteries, bringing about capacity degradation, internal short circuits and even thermal runaway [15,16]. Therefore, modelling lithium plating evolution with various working conditions through reaction kinetics that really matters. The prevention of lithium evolution can be achieved by implanting the element above into the constraints of the system. The Butler–Volmer equation as shown in Equation (13) is generally recognized as a tool for lithium plating description.

$$j_{Li}(t) = min\left\{0, j_{0,Li}\left[\exp\left(\frac{\alpha_a F}{RT_{bat}}\eta_{Li}\right) - \exp\left(-\frac{\alpha_c F}{RT_{bat}}\eta_{Li}\right)\right]\right\} \tag{13}$$

where $j_{0,Li}$ is the exchange current density of the lithium plating reaction, $\alpha_a$ and $\alpha_c$ are the dimensionless anodic and cathodic charge transfer coefficients, $\eta_{Li}$ is the overpotential of lithium plating side reaction defined as

$$\eta_{Li} = \varphi_{s,n} - \varphi_e - U_{Li}^{ref} - R_{SEI}i_{Li} \tag{14}$$

$\varphi_{s,n}$ and $\varphi_e$ are the potentials of the solid phase and the electrolyte, respectively, $U_{Li}^{ref}$ is the equilibrium potential of the lithium deposition side reaction, which is set to be 0V. It is worth noting that only when $\varphi_{s,n}$ is lower than $\varphi_e$, Lithium plating occurs. $\varphi_{s,n} - \varphi_e$ could be derived from the overpotential of the main reaction as

$$\varphi_{s,n} - \varphi_e = \eta_m + U_n^{ref} + \frac{R_{SEI}I}{\varepsilon_{s,n}A\delta_n} \tag{15}$$

$\eta_m$ has been accounted in Equations (5) and (6). $U_n^{ref}$ is relevant with battery SOC and $R_{SEI}$ is calibrated considering the temperature effect of lithium deposition under the Arrhenius type correlation. $\varepsilon_{s,n}$ is the volume fraction of the solid phase of the negative particle, which decreases with the loss of active material. It deserves to emphasize that in real batteries, there exists an onset potential $\varphi_{onset}$ that lithium plating will not happen until $\varphi_{s,n} - \varphi_e$ is lower than $\varphi_{onset}$. As a result, the reduce-order lithium plating model is summarized as Equation (16), where $k_{Li}$ and $\varphi_{onset}$ are parameters to be calibrated. Besides, $\Phi$ describes the relationship between the overpotential with the lithium plating current density.

$$i_{Li}(t) = k_{Li}A\varepsilon_{s,n}\delta_n\Phi\left(\eta_m + U_n^{ref} + \frac{R_{SEI}I}{A\varepsilon_{s,n}\delta_n} - \varphi_{onset}\right) \tag{16}$$

### 2.4. Coupling Principle for Sub-Models

It is critical to couple the three sub-models for the connection with capacity. Taking into account that the side reactions of lithium-ion batteries are chained, the three sub-models proposed should be coupled with each other. The reduced-order physical degradation model with the coupling principle is shown in Figure 2. The pivotal coupling parameters are indicated by the red dashed line.

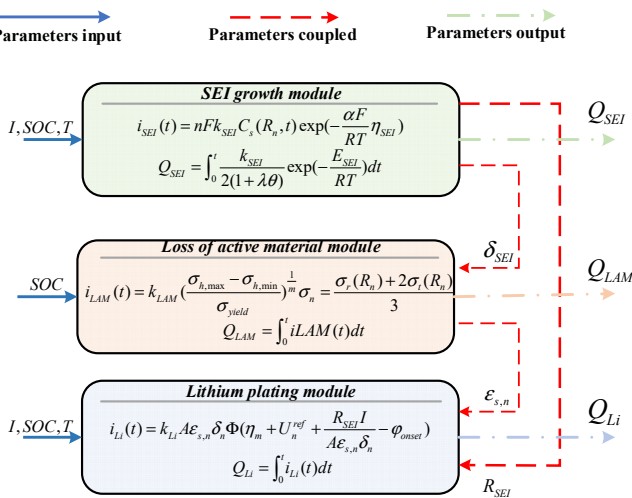

**Figure 2.** Degradation model with coupling principle.

It is known that SEI growth promotes LAM and lithium plating. SEI film layer radius $R_s$ in the LAM increases with $\delta_{SEI}$, further affecting the lithium plating rate. The reduced volume fraction $\varepsilon_{s,n}$ also accelerates the lithium deposition rate of on the anode. Eventually, the consequence of chain side effects will be reflected in capacity degradation. The damage coefficients of each sub-module in Table 1 wait to be calibrated, which will be illustrated in the next section.

**Table 1.** Calibrated parameters.

| Parameters | Value | Parameters | Value |
|---|---|---|---|
| $\lambda$ | 86,995 | $\sigma_{yield}$ | 787.41 MPa |
| $k_{SEI}$ | 15.55 s$^{-1/2}$ | $m$ | 0.23 |
| $E_{SEI}$ | 27,219 J/mol | $k_{Li}$ | 4.20 |
| $k_{LAM}$ | 1.61 | $\varphi_{onset}$ | −5.5 mV |

## 3. Calibration and Verification

### 3.1. Microgrid Operations Simplified for Accelerated Degradation

When the physical degradation model is implemented in evaluating the battery health status with energy storage conditions, it is essential to consider the applicability under the practical operations. For this reason, the aging experiments carried out in this paper are designed based on real microgrid conditions, which are obtained through photovoltaic-based DC microgrids for fast charging stations for EV [17]. Due to the fluctuant current, it is not conducive for the control in laboratory. Taking the equivalent conditions, the 12-hour operating conditions are equivalently simplified and smoothed into multiple constant currents as shown in Figure 3a. In addition, five increasingly severe accelerations are applied to five graphite-LiFePO4 batteries with initial nominal capacity of 20 Ah in a chamber of 21 °C for the aging performance at different C-rates and SOC ranges. As shown in Figure 3b,c, the current and SOC are specifically depicted. Obviously, the microgrid working condition simplified takes 144 min as a cycle, and the number of cycles will be selected as the basic unit in the following graphs.

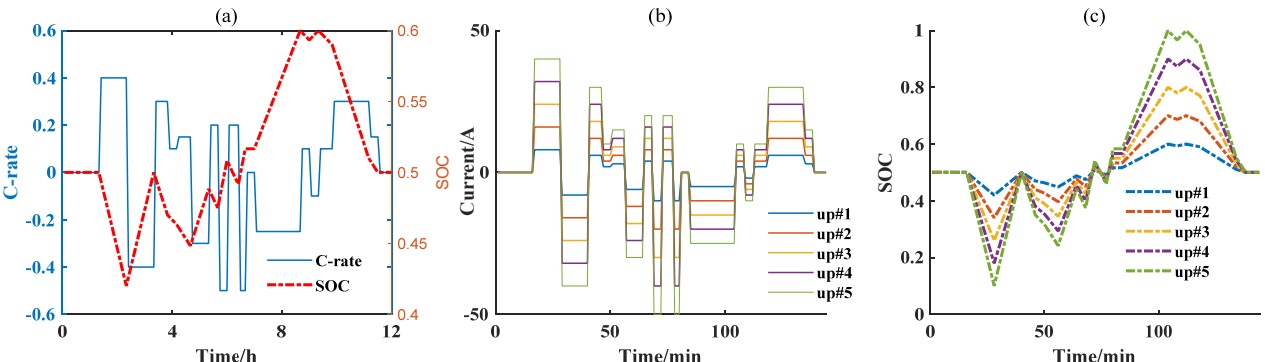

**Figure 3.** Degradation speeded-up tests simplified. (**a**) Microgrid operations simplified equivalently; (**b**) Current under different C-rates; (**c**) SOC under different C-rates.

### 3.2. Result Analysis and Model Verification

With the five operations in Figure 3b,c as the base, 500 cycles are carried out for the five batteries, of which a standard capacity test (SCT) is performed every 50 cycles. The accelerated capacity degradation are expressed in Figure 4a. It is seen that as the C-rates increases, the capacity ages more significantly, and the overall degradation presents a linear trend. It is worth noting that the battery capacity with up #5 operation is drastically reduced due to harsh conditions, and the cycles are limited to 400 times only (fails to be charged efficiently). As the C-rate and SOC range increase, the capacity degradation becomes pronounced. The available capacity drops sharply after 300 cycles especially under the up #5 working conditions. In particular, the degradation in the later stage shows a nonlinear trend.

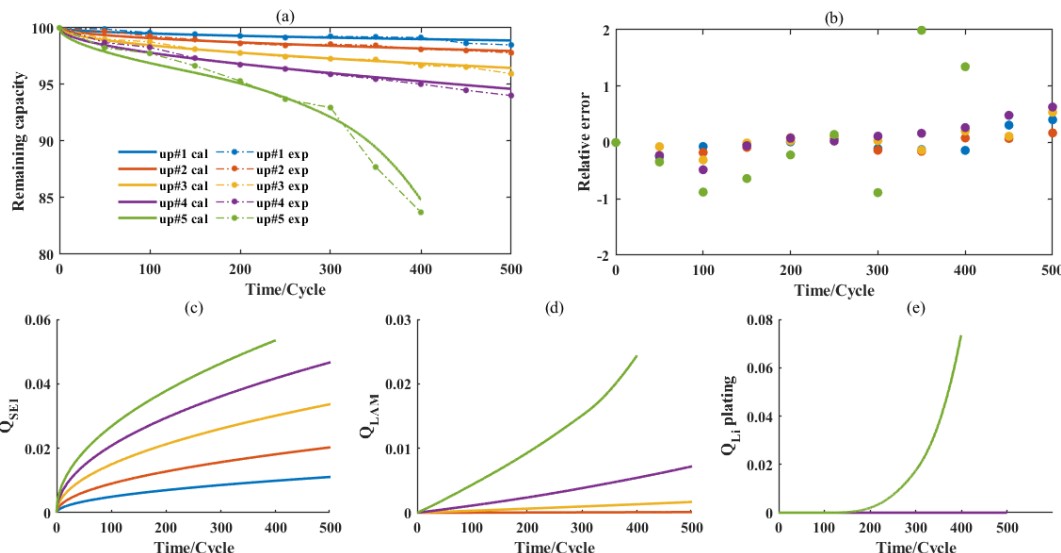

**Figure 4.** Calculation result of degradation model. (**a**) Display of results between experiments and calculations; (**b**) Relative error; (**c**). Capacity loss brought by SEI growth sub-module; (**d**) Capacity loss brought by LAM sub-module; (**e**) Capacity loss brought by lithium plating sub-module.

The degradation constructed by the three sub-models is a nonlinear and strongly coupled process. An effective optimization algorithm is essential for the complex issues. Therefore, the intelligent particle swarm optimization (PSO) algorithm with few parameter settings and strong optimization ability is employed here [18], obtaining optimal parameters of the three models under five groups of working conditions. Relying on cycling life tests, the parameters are calibrated for the proposed reduced-order degradation model. The calibrated and other physical parameters are presented in Tables 1 and 2, respectively.

**Table 2.** Physical parameters.

| Parameters | Value | Parameters | Value | Parameters | Value |
|---|---|---|---|---|---|
| $n$ | 1 | $\delta_n$ | 16.7 μm | $E_p$ | 15 Gpa |
| $F$ | 96,487 | $\varepsilon_{s,n0}$ | 0.59 | $E_s$ | 0.5 Gpa |
| $\alpha$ | 0.5 | $\delta_{SEI0}$ | 0.02 μm | $v_p$ | 0.3 |
| $R$ | 8.314 | $R_{SEI0}$ | 4 mΩ | $v_s$ | 0.2 |
| $R_n$ | 9 μm | $M$ | 0.162 kg/mol | $c_{p,max}$ | 31.92 mol/dm$^3$ |
| $i_0$ | 2 A/m$^2$ | $\rho$ | 1690 kg/m$^3$ | $\Omega_p(c_p, max)$ | 3.1 cm$^3$/mol |
| $A$ | 1 m$^2$ | $\sigma_{SEI}$ | $5 \times 10^{-6}$ S/m | $U_{SEI}^{ref}$ | 0.4 V |

The temperature, current and SOC are input into the three sub-models proposed in the Section 2, and the respective capacity losses are obtained in Figure 4c–e. It is shown that the capacity damage of SEI and LAM increases with C-rate and SOC range. Among them, the damage caused by SEI growth is more significant than that of LAM. The lithium plating in Figure 4e is worth noting. In fact, the battery with up #1~up #4 has not undergone lithium plating, but once it occurs such as up #5, significant capacity degradation will happen. Lithium plating is the main cause for the sudden capacity decrease (non-linear degradation) in the fifth battery. According to the simulation results, the reduced-order physical degradation model is in good agreement with the experiments as depicted in Figure 4a and the relative error of the five operations is kept within 2%. Both linear and nonlinear degradation can be revealed, especially the nonlinear degradation with up #5 operation. The degradation model proposed therefore is credible to achieve fine accuracy.

## 4. Dual Closed-Loops Capacity Prediction Framework

The dual closed-loops capacity prediction framework is advanced to solve the closed-loop mechanism interpretation and the closed-loop parameter updating for capacity prediction. On the one hand, the calculation of the side reactions in Section 3 ought to be further qualitatively verified; on the other hand, the physical degradation model is only applied to capacity estimation but not prediction due to the unknown future working conditions. Capacity prediction is able to be realized in an empirical model, and the parameters should not be static, but to be updated within the full life. Therefore, the closed-loop updating for parameters really matters and the dual closed-loops capacity prediction framework shown in Figure 5 is proposed to address the problems above.

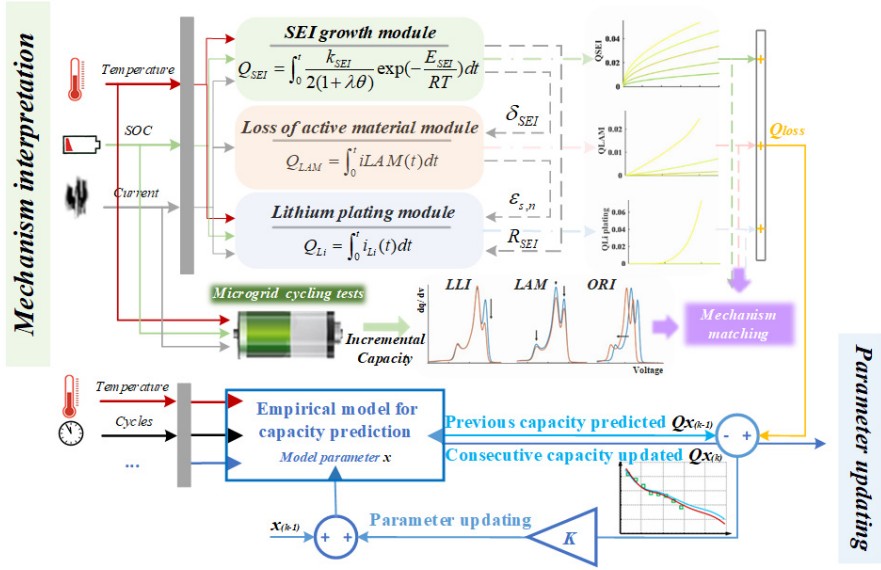

**Figure 5.** Dual closed-loops capacity prediction framework.

### 4.1. The First Closed Loop: Mechanism Interpretation

Incremental capacity analysis (ICA) has recently been utilized as a recognized technique to evaluate the state inside [19], especially for state of health. In this section, the calculation of the physical degradation sub-model will be non-destructively verified via ICA to obtain LLI, LAM, and ORI as shown in Figure 5. It can be analyzed quantitatively from the perspective of a more rigorous mechanism, but the qualitative analysis with a higher priority based on ICA is taken into application as a rough verification in this paper. Level evaluation analysis (LEAN) is adopted for the acquisition of IC curves every 100 cycles [20]. By dividing the voltage sampling interval and recording the number of sampling points in each interval, the authenticity of IC curve is guaranteed. IC curves of the five experiments at different aging processes is drawn in Figure 6, where (a)~(e) correspond to up #1~up #5 respectively.

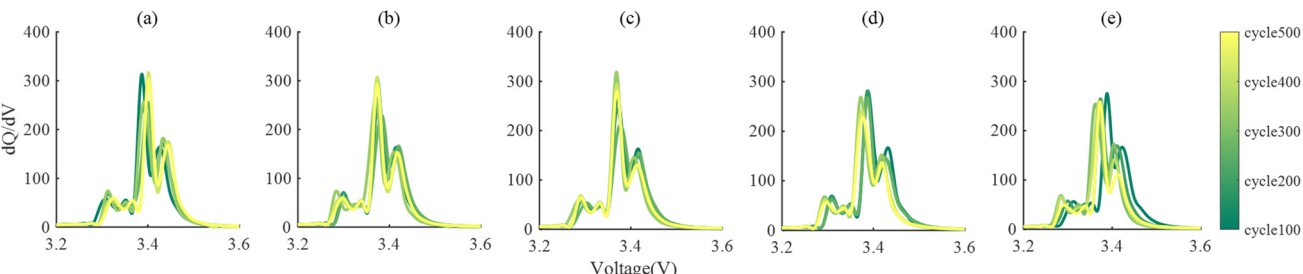

**Figure 6.** Incremental capacity analysis. (**a**) Incremental capacity evolution with up#1; (**b**) Incremental capacity evolution with up#2; (**c**). Incremental capacity evolution with up#3; (**d**) Incremental capacity evolution with up#4; (**e**) Incremental capacity evolution with up#5.

IC curve reflects the degradation mechanism. It is clearly observed in Figure 6 that as the operations applied become more serious, the third peak (from left to right) of the IC in (a)~(e) decreases more obviously. The increasingly serious internal loss of lithium inventory (LLI) is reflected, which matches the growth of SEI and lithium plating in Figure 4, because SEI growth and lithium plating are responsible for LLI. In addition, LAM is also verified through ICA. The overall height of the three peaks indicates LAM and it does not significantly change from Figure 6a–c. The decrease occurs in (d) and (e), which also matches the calculation for LAM in Figure 4d. The shift of the IC curve in Figure 6e is also worthy of attention, which presents the growth of internal resistance. The mechanism is also consistent with the lithium plating in Figure 4e. As the amount of lithium plating accumulates, the internal resistance of the battery gradually increases. Based on the qualitative analysis above-mentioned, it is confirmed that the model proposed matches the degradation mechanism truly experienced by the actual battery. The first closed loop for the mechanism interpretation is constructed.

### 4.2. Second Closed Loop: Parameter Updating

The current, SOC, and temperature are imported as the input of the physical reduced-order degradation model for SOH evaluation. This degradation model is limited to present estimates and not future predictions. The prediction for future degradation trends often adopts empirical models, but empirical models with constant parameters are considered unreasonable due to parameter mismatch caused by complex operating conditions. Therefore, the second closed loop is proposed as the lower part of Figure 5 for parameter updating and its principle lies in state estimation. Considering that the physical model provides precise estimates in Section 3, it is employed as a reliable observation to modify the parameters throughout the life cycle. The parameters revised are brought into the empirical model to predict capacity evolution.

Features are worth considering in the empirical model selection and the models applied to distinct side reactions show concave or convex characteristics. It is well known



that the capacity loss of lithium-ion batteries is usually convex due to growth of SEI on the anode graphite surface [21]. However, when the battery reaches the end of life, degradation rate increases significantly. The degradation trend will be concave and verified in up #5 of Figure 4a. Therefore, the selection of the empirical model relies on both the aging process and working condition. In this paper, the empirical model corresponding to up #1–up #4 ought to utilize convex function; in the case of up #5, the double exponential model with concave characteristics is adopted for prediction considering the nonlinear degradation caused by lithium plating as follows [22]. It is worth clarifying that the first four operating conditions are basically linear, and there is no major challenge in predicting the evolution of their capacity. This article focuses on the prediction of the non-linear capacity degradation in the case of up #5. If the capacity evolution process accompanying lithium plating can be accurately predicted, the significance of parameter updating will be verified.

$$Q = a * \exp(b * n) + c * \exp(d * n) \tag{17}$$

where $Q$ is the battery capacity, $n$ is the cycling number, and $a, b, c, d$ are system state estimated. The state equations for the prediction model are

$$X_n = [a_n \ b_n \ c_n \ d_n]^T \tag{18}$$

$$X_{n+1} = X_n + w_n, \ w \sim N(0, \sigma_w) \tag{19}$$

The observation equation is expressed in Equation (20), $w$ and $v$ are Gaussian white noise.

$$Q_n = a_n \exp(b_n * n) + c_n \exp(d_n * n) + v_n, v \sim N(0, \sigma_v) \tag{20}$$

Particle filtering (PF) based on Bayesian filtering and Monte Carlo sampling is applied in this paper during state estimation [23]. System state $X_n$ is estimated through posterior probability density function (PDF) $p\{X_n|Q_{1:n}\}$. The general steps are elaborated as follows. Initialization ought to be performed first when $n = 0$, the particles $\left\{ \left( X_0^{(i)}, \omega_0^{(i)} \right) \right\}_{i=1}^{N}$ are sampled from initial distribution $p(X_0)$. $N$ is the number of particles configured as 100, the weight of each particle is given initially the same as $\omega_0^{(i)} = \frac{1}{N}$. Sequential importance sampling (SIS) follows. Relying on the importance PDF, the particle sets $\left\{ X_n^{(i)} \right\}_{i=1}^{N}, i = 1, 2, \ldots, N$ are sampled randomly and generated.

$$X_n^{(i)} \sim q\left( X_n^{(i)}|X_{n-1}^{(i)}, Q_{1:n} \right) = p(X_n^{(i)}|X_{n-1}^{(i)}) \tag{21}$$

where $X_n^{(i)}$ represents the $i$-th particle with n-th sequence; $Q_{1:n}$ denotes the observed value resulting from the capacity calculation of coupled electrochemical degradation model presented; $q(X_n^{(i)}|X_{n-1}^{(i)}, Q_{1:n})$ is the importance probability density function and $p(X_n^{(i)}|X_{n-1}^{(i)})$ is the prior probability.

Then the weight of each particle at each moment is going to be calculated. Generally, $p(X_n|X_{n-1})$ is ordinarily selected as the importance probability density function when calculating the weight, satisfying Equation (22).

$$\omega_n^{(i)} = \omega_{n-1}^{(i)} \frac{p(Q_n|X_n^{(i)}) \cdot p(X_n^{(i)}|X_{n-1}^{(i)})}{q(X_n^{(i)}|X_{n-1}^{(i)}, Q_{1:n})} \propto \omega_{n-1}^{(i)} \cdot p(Q_n|X_n^{(i)}) \tag{22}$$

where $\omega_n^{(i)}$ represents the weight value of the $i$-th particle at time $n$, and '$\propto$' means 'being proportional to'. The weight is incidentally normalized as

$$\omega_n^{(i)\prime} = \frac{\omega_n^{(i)}}{\sum_{i=1}^{N} \omega_n^{(i)}} \tag{23}$$

It is worth noting that SIS is vulnerable to particle degeneracy. After many iterations, the particles become concentrated that numerous particles contribute little due to its small weights, which has been proved to be inevitable [24]. Hence the effective sample size $N_{eff}$ is evaluated.

$$N_{eff} = \frac{1}{\sum_{i=1}^{N} \left( \omega_n^{(i)'} \right)^2} \tag{24}$$

The smaller the $N_{eff}$ is, the more degeneracy the particles indicate. Resampling is thus advanced to cope with the particle degeneracy [25]. It is generally believed that resampling is performed when $N_{eff} < \frac{2}{3}N$. By introducing resampling, the posterior PDF with n-th sequence is approximated as

$$p(X_n|Q_{1:n}) \approx \sum_{i=1}^{N} \omega_n^i \delta \left( X_n - X_n^{(i)} \right) \tag{25}$$

where $\delta(\cdot)$ represents Dirac delta function. The empirical model parameters with k-th sequence are estimated by Equation (26) eventually.

$$\hat{X}_n = \sum_{i=1}^{N} X_n^{(i)} \cdot \omega_n^{(i)} \tag{26}$$

During the prediction verification for lithium plating condition, the entire experimental process is converted into hourly scale (400 cycles × 144 min/60 min = 960 h). The capacity calculation of the coupling degradation model in the first 560 h is taken as an observational reference to predict capacity evolution in the next 400 h. The reason for this operation is that the calculation in the previous stage through coupling degradation model includes the non-linear degradation message brought by lithium plating in Figure 4e. Relying on instantaneous state estimation, the state system is capable of sensitively capturing tiny degradation messages implicit in observations.

Taking 560 h as beginning for prediction, capacity evolution for the next 400 h is predicted as depicted in Figure 7. In the early stage, due to a certain error between initial parameters and state estimated, the capacity is in a fluctuating state for parameter updating. With the consequent output of coupling degradation model, the parameters of empirical model are updated all the time in Figure 7a–d. It is deemed that the true degradation trend has basically been approached at 560 h, so the remaining capacity prediction is carried out. The parameters updated finally are employed as the input of empirical model for future prediction. According to the prediction, the degradation trend is basically consistent with experiments.

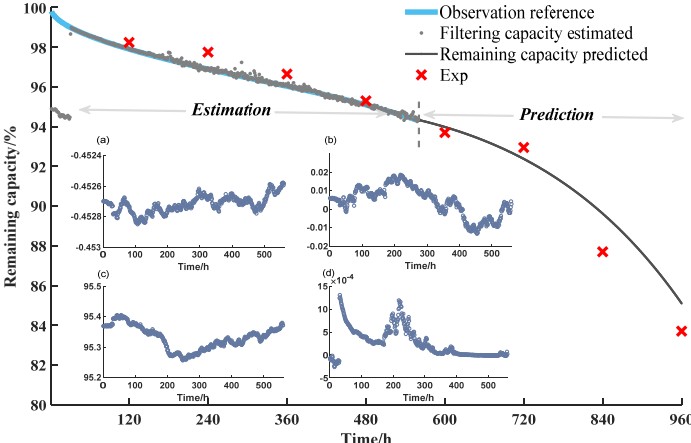

**Figure 7.** State estimation for capacity prediction. (**a**) Evolution of $a_n$ updated; (**b**) Evolution of $b_n$ updated; (**c**). Evolution of $c_n$ updated; (**d**) Evolution of $d_n$ updated.

## 5. Conclusions

The dual closed-loops capacity framework is proposed for the mechanism explanation and parameter updating of energy storage batteries. On the one hand, a coupled degradation model containing three side reactions SEI growth, LAM, and lithium plating is developed to accurately assess health status from a mechanism perspective. At the same time, the real microgrid working conditions simplified are implemented for the first time to the validity of the model. In addition, in order to test whether the model calculation reflects the real degradation history, the first closed loop mechanism explanation based on ICA technology is advanced. It is recognized that the qualitative results relying on ICA are basically consistent with the model calculation. On the other hand, for the purpose of degradation evolution prediction, the second closed loop parameter updating follows one after another. Taking into account the capacity output of the coupled degradation model with high accuracy, it is appropriate to provide reliable observations for parameter updating of empirical model through state estimation algorithm. In the ultimate, it is verified with the battery accompanied by lithium plating, and it is affirmed that the double exponential empirical model with parameters updated is capable of basically revealing the non-linear capacity evolution trend resulting from lithium plating.

The framework coupled with aging mechanism drives the update for empirical parameters, which not only reveals the side effects dominated at each degradation stage, but also accurately predicts its capacity evolution with error kept within 2%, making up for the lack of mechanism support in both the general empirical and data-driven black box models for state estimation. It is worth noting that the degradation mechanism is currently only qualitatively analyzed. The evolution of various side reactions can be roughly grasped relying on ICA, and the analysis results can basically match the calculation results. In subsequent research, more specific and quantitative electrochemical mechanism technology will be considered, and the capacity loss caused by SEI, LAM, and lithium plating in Figure 4 will be accurately verified, which helps the reliability of the coupled degradation model to be applied to actual energy storage.

**Author Contributions:** Conceptualization, B.X., S.W., Y.W. and Y.Z.; Methodology, B.X., S.W. and Y.Z.; Software, B.X., S.W. and Y.W.; Validation, B.X., S.W. and Y.W.; Formal analysis, S.W. and Y.W.; Investigation, B.X. and S.W.; Resources, T.S., X.H. and Y.Z.; Data curation, S.W. and Y.W.; Writing—original draft preparation, B.X. and S.W.; Writing—review and editing, B.X.; Visualization, B.X.; Supervision, T.S., X.H. and Y.Z.; Project administration, B.X.; Funding acquisition, T.S., X.H. and Y.Z. All authors have read and agreed to the published version of the manuscript.

**Funding:** This research was funded by International Science & Technology Cooperation of China, grant number 2019YFE0100200; Tsinghua-Toyota Joint Research Institute Cross-discipline Program, the National Natural Science Foundation of China (NSFC), grant number 51877138; and Shanghai Science and Technology Development Fund, grant number 19QA1406200.

**Conflicts of Interest:** The authors declare no conflict of interest. The funders had no role in the design of the study; in the collection, analyses, or interpretation of data; in the writing of the manuscript, or in the decision to publish the result.

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
