# Peer review of "Dual Closed-Loops Capacity Evolution Prediction for Energy Storage Batteries Integrated with Coupled Electrochemical Model"

_wevj, doi:10.3390/wevj12030109_

Round 1
Reviewer 1 Report
This is the review of the manuscript entitled „Dual closed-loops capacity evolution prediction for energy storage batteries integrated with coupled electrochemical model”.
The authors present an interesting topic, being in line with the WEVJ mission.
In general, this manuscript is well organized and written, with a comprehensive literature review, detailing the framework approach of the study, clearly stated methodology, and nicely presented findings. The manuscript provides sufficient background information regarding the topic proposed.
Suggestion:
The conclusion section is missing some perspective related to the future research work, quantify main research findings.
- provide sharper Figure 2.
Reviewer 2 Report
1. Briefly summarize the content of the manuscript;
The manuscript proposes a dual closed-loop capacity prediction framework able to update aging parameters of energy storage batteries.
2. Illustrate what are, in your opinion, the manuscript’s strengths and weaknesses [this is an essential step, because the Editor will consider the reasoning behind your recommendation and needs to understand it properly];
strengths - detailed explanations of the mathematical/physical/chemical equations that lead to the proposed framework
weaknesses - none, imho
3. Provide a point-by-point list of your scientific recommendations for the improvement of the manuscript, apart from the spelling/formatting errors;
---
4. If necessary, provide a point-by-point list of your minor for the improvement of the manuscript.
figure 3, 4 some of the colors are too light to be seen, consider replacing the color map
